# Enhancing Nurse Practitioners’ Emergency Care Competency and Self-Efficacy Through Experiential Learning: A Single-Group Repeated Measures Study [note 1]

**DOI:** 10.3390/healthcare12232333

**Published:** 2024-11-22

**Authors:** Ya-Lun Yang, Li-Chuan Cheng, Chen-Wei Lee, Shih-Chun Lin, Malcolm Koo

**Affiliations:** 1Department of Nursing, Dalin Tzu Chi Hospital, Buddhist Tzu Chi Medical Foundation, Dalin 62247, Taiwan; 2Department of Emergency, Dalin Tzu Chi Hospital, Buddhist Tzu Chi Medical Foundation, Dalin 62247, Taiwan; mike121280mike@gmail.com; 3Graduate Institute of Nursing, National Taipei University of Nursing and Health Sciences, Taipei City 112303, Taiwan; 4Department of Nursing, Tzu Chi University, Hualien City 970302, Taiwan

**Keywords:** nurse practitioners, self-efficacy, experiential learning theory, competencies, Taiwan

## Abstract

Background/objective: Nurse practitioners serve a vital role as first responders in emergencies. This study investigated the effectiveness of experiential learning in enhancing emergency care competency and self-efficacy among nurse practitioners. Methods: A single-group repeated measures design was implemented from June to August 2023 at a regional teaching hospital in southern Taiwan, involving 95 nurse practitioners and NP trainees. Participants completed a baseline (T_0_) three-minute emergency simulation test, followed by one-on-one guidance, an immediate post-test (T_1_), and a follow-up test one month later (T_2_). The “Emergency Care Capability Checklist” (ECCC) was used to assess performance after each test, and the “General Self-Efficacy Scale” at T_1_ and T_2_. Results: The mean age of the participants was 42.1 years (SD = 6.7), with 91 out of 95 participants (95.8%) being female. ECCC scores increased significantly from a baseline mean of 34.6 (standard deviation [SD] = 8.8 at T_0_ to 46.4 (SD = 4.3) at T_1_ (*p* < 0.001). Scores remained elevated at T_2_, with a mean of 44.7 (SD = 4.9), which was significantly higher than T_0_ (*p* < 0.001). However, scores at T_2_ were slightly lower than at T_1_ (*p* = 0.018). GSES scores also increased significantly from T_1_ (mean = 26.2, SD = 0.6) to T_2_ (mean = 28.0, SD = 0.6) (*p* = 0.009). Conclusions: This study found that experiential learning was able to significantly improve nurse practitioners’ emergency care competencies and self-efficacy. Future research should explore the application of experiential learning in diverse clinical settings to further advance emergency preparedness and self-efficacy among nurse practitioners.

## 1. Introduction

In the rapidly evolving healthcare landscape, nurse practitioners play an increasingly critical role. As advanced practitioners within the nursing profession, nurse practitioners manage both acute and chronic medical conditions, coordinate care across specialties, and provide continuous, integrated care. Their emergency response skills are particularly essential, as they often serve as frontline responders in critical situations [1]. In Taiwan, regional and local hospitals often experience a shortage of resident physicians, who are typically concentrated in medical centers. As a result, nurse practitioners frequently serve as first-line responders in inpatient wards during night shifts or weekends, performing initial assessments when immediate medical support is limited. In these settings, when a patient’s condition deteriorates or vital signs become unstable, the primary nursing staff first contacts a nurse practitioner, who conducts an initial assessment and stabilizes the patient before determining whether additional physician support is needed. If cardiac or respiratory arrest occurs, a hospital-wide code call will be initiated to activate the cardiopulmonary resuscitation team, ensuring that the full emergency response team arrives promptly. In addition, nurse practitioners can provide adjunctive care under a physician’s guidance, including ordering and interpreting diagnostic tests and initiating basic treatment protocols. This scope of practice enables nurse practitioners to effectively manage critical situations as primary responders and to coordinate with the clinical team, especially in urgent situations when physicians are not immediately available.

The effectiveness of nurse practitioners in emergency scenarios depends on two key factors: their clinical competency and their confidence in executing these skills under high-pressure conditions. This progression of clinical competence can be explained by Patricia Benner’s novice to expert model, which describes five levels of skill acquisition in nursing practice: novice, advanced beginner, competent, proficient, and expert [2]. Benner’s model provides a theoretical foundation for understanding the development of nurse practitioners as they gain experience, allowing them to handle complex scenarios with greater autonomy and confidence. This intersection of skill and confidence aligns with two fundamental concepts in medical education—experiential learning and self-efficacy [3]—providing a foundation for understanding and enhancing the performance of nurse practitioners in acute care settings.

Experiential learning theory, proposed by David Kolb in 1984, emphasizes the importance of learning through hands-on experience and reflection. The theory outlines four stages: concrete experience, reflective observation, abstract conceptualization, and active experimentation. This cyclical process encourages learners to actively participate in their education, fostering deeper understanding and retention of knowledge [4]. A review study indicated that experiential learning, a learner-centric approach, utilizes real-world scenarios to enhance engagement and active learning among students. By linking theory to practice, this method not only improves practical skills but also strengthens intrinsic motivation by making learning more relatable and interactive [5]. In medical education, where theoretical knowledge must be rapidly applied in high-stress environments, this approach is particularly relevant [6].

Confidence in one’s abilities relates to the concept of self-efficacy, introduced by Bandura in 1977. Self-efficacy refers to an individual’s belief in their ability to execute specific tasks successfully [7]. In the context of emergency care, high self-efficacy is vital. Nurse practitioners with strong self-efficacy are better equipped to manage crisis situations and improve patient outcomes [8].

Experiential learning has been shown to significantly enhance self-efficacy. In one study using a pre-test/post-test comparison group design, Doctor of Physical Therapy students who participated in pediatric experiential learning demonstrated improvements in both clinical reasoning and self-efficacy [9]. Similarly, a quasi-experimental study with a pre- and post-test design involving 249 nursing students and 50 medical students found that experiential learning in a mental health module improved their empathy and confidence in managing dangerous, aggressive, and violent patients [10]. Another quasi-experimental study, involving 285 nursing students who participated in simulated practice programs, showed significant improvements in self-confidence for emergency intervention, which translated into enhanced knowledge and practical skills [11]. A single-group pre- and post-comparison study of pediatric emergency physicians also reported greater confidence and perceived ability to manage patients in a disaster situation following experiential learning through simulation-based workshops [12]. Collectively, these studies support the effectiveness of experiential learning in enhancing self-efficacy and confidence across diverse clinical settings, supporting its value as a powerful educational approach for improving healthcare professionals’ readiness to manage complex and dynamic healthcare environments.

The critical role of nurse practitioners in emergency settings, particularly during holidays or night shifts when they may be the first responders, shows the importance of effective training. In the critical first three minutes before the arrival of an emergency response team, the functional capabilities of nurse practitioners are paramount [13]. Therefore, training strategies that simultaneously improve clinical skills and self-efficacy are essential.

Given the evidence supporting the benefits of experiential learning and its role in enhancing self-efficacy, there is a clear need to further explore its application in nurse practitioner training, particularly in emergency care. Therefore, this study aimed to evaluate the effectiveness of experiential learning in improving both emergency care competency and self-efficacy among nurse practitioners.

## 2. Materials and Methods

### 2.1. Study Design and Participants

The study protocol was approved by the Research Ethics Committee of Dalin Tzu Chi Hospital, Buddhist Tzu Chi Medical Foundation (No. B11202019). All participants provided informed consent prior to the study.

This study employed a single-group repeated measures design, conducted between June and August 2023. Participants included nurse practitioners and nurse practitioner trainees from a regional teaching hospital in southern Taiwan.

### 2.2. Testing Sessions

The study involved three test sessions: a baseline session (T_0_), an immediate post-intervention session (T_1_), and a follow-up session one month later (T_2_). During each session, participants completed a test and were assessed on their emergency care skills and self-efficacy.

### 2.3. Baseline and Post-Intervention Procedures

At the baseline session (T_0_), participants completed an initial test followed by one-on-one guidance from examiners, who demonstrated correct emergency care procedures via a laptop and engaged in feedback. Participants completed the second test session (T_1_) immediately after this baseline intervention. One month later, participants returned for the follow-up test (T_2_), where the same procedures were repeated.

### 2.4. Assessment Tools and Scenarios

In each session, participants were evaluated using the Emergency Care Competency Checklist (ECCC) and the General Self-Efficacy Scale (GSES). Participants were presented with a scenario in which a patient admitted with unstable angina became unresponsive during a blood pressure check. Participants were required to use the call bell to summon assistance and interact with three standardized nurses. Their clinical tasks included performing emergency procedures, problem-solving, operating an automated external defibrillator (AED), and communicating effectively. Examiners observed and recorded the participants’ performance using the ECCC. The duration of each test was 3 min. After completing the test, participants filled out the GSES. In addition, examiners played a video on a laptop computer demonstrating the correct emergency care procedures and provided one-on-one feedback and guidance.

### 2.5. Examiner and Standardized Nurse Training

The examiner overseeing the tests was a certified nurse practitioner with over 8 years of critical care experience who underwent 4 h of training with attending physicians, along with two assessments to ensure consistency and accuracy in scoring. Three standardized nurses, trained to follow a scripted protocol, participated in the scenarios. Prior to the baseline test, these standardized nurses attended a one-hour training led by an attending physician and a Department of Medical Education staff member, followed by a review session on the day of the examination to ensure nurse preparedness.

On the day of the examination, the attending physician conducted an additional one-hour review to ensure the standardized nurses were adequately prepared. The tests were conducted in a simulated ward environment within the study hospital, an accredited national examination site recognized by Taiwan’s Ministry of Health and Welfare.

### 2.6. Measurement Scales

The ECCC was designed through a series of collaborative meetings with attending physicians and emergency department attending physicians at the study hospital. The checklist’s content and structure were guided by a quality management analysis of emergency response incidents within our hospital, as well as the 2020 American Heart Association Guidelines Update for Cardiopulmonary Resuscitation and Emergency Cardiovascular Care [14]. It encompasses essential emergency care competencies alongside common clinical errors observed in our setting, aiming to reflect both critical competencies and the practical challenges of emergency care. The ECCC consists of 26 items, each rated on a three-point scale: “not achieved” (1 point), “partially achieved” (2 points), and “fully achieved” (3 points) (Appendix A).

The General Self-Efficacy Scale (GSES) was selected for this study due to its proven reliability, validity, and wide applicability in assessing self-efficacy across diverse life challenges. The GSES is a widely validated, unidimensional self-reported measure to assess an individual’s level of self-efficacy in coping with various life challenges. Originally developed by Jerusalem and Schwarzer [15], the GSES has been adapted into more than 30 languages [16]. It includes 10 items rated on a 4-point Likert scale, with responses ranging from “not at all true” to “exactly true”. Total scores range from 10 to 40, with higher scores reflecting greater self-efficacy.

The GSES has demonstrated good internal consistency and reliability across different research and cultural contexts [17,18]. In a large-scale study involving 19,120 participants from 25 countries, Cronbach’s alphas ranged from 0.75 to 0.91 with an overall alpha of 0.86 [19]. In addition, GSES shows good criterion-related validity, as evidenced by studies linking higher GSES scores with positive outcomes, such as greater life satisfaction [20], improved coping skills [21], and enhanced stress management [22]. Moreover, in a cross-cultural study spanning 25 countries, the GSES demonstrated positive correlations with optimism and expected social support, and inverse correlations with anxiety and depression, further supporting its criterion-related validity as a measure of self-efficacy [23].

The Chinese version of the GSES was employed in this study. A Chinese adaptation study conducted on 293 university students reported an internal consistency Cronbach’s alpha of 0.91 [24]. Another study on university students using principal component analysis showed that the Chinese version is unidimensional, with the first factor accounting for 55% of the variance, supporting the scale’s psychometric robustness and suitability for further research [25].

### 2.7. Data Analysis

Continuous variables were expressed as means and standard deviations (SDs), while categorical variables were presented as frequencies and percentages. Paired-sample *t*-tests were used to compare ECCC across T_0_, T_1_, and T_2_, as well as GSES scores between T_1_ and T_2_. To account for multiple comparisons, the Bonferroni correction was applied to adjust *p*-values. Pearson’s correlation coefficient was calculated to assess the relationship between the change in GSES scores from T_1_ to T_2_ and the change in ECCC scores from T_0_ to T_1_. All statistical analyses were conducted using IBM SPSS Statistics for Windows, version 26.0 (IBM Corp., Armonk, NY, USA).

## 3. Results

A total of 95 nurse practitioners and nurse practitioner trainees participated in the study, with no dropouts. The mean age of the participants was 42.1 years (SD = 6.7 years). Of these, 91 (95.8%) were female, 89 (93.7%) were nurse practitioners, and 68 (71.6%) had completed a university-level education. The largest department represented was internal medicine, with 31 participants (32.6%). In addition, 90 participants (94.7%) held emergency-care-training-related certifications.

In terms of skill level, 63 participants (66.3%) were at the NPII level. In Taiwan, nurse practitioners advance through a clinical ladder and certification system that supports career progression based on increasing clinical competence and responsibility. Starting at the trainee level, N represents nurse practitioner trainees who are preparing for clinical roles. From there, the levels range from NPI to NPV as follows: (1) NPI—novice: becoming familiar with the nurse practitioner scope of practice; (2) NPII—advanced beginner: capable of handling basic nurse practitioner duties and providing individualized patient care; (3) NPIII—competent: able to develop evidence-based care plans and contribute opinions within the medical team; (4) NPIV—proficient: manages overall patient conditions, applies research to care, and coordinates with the medical team; and (5) NPV—expert: makes rapid, accurate clinical judgments, integrates research, and influences the medical team and institution (see Table 1).

ECCC scores showed a significant increase, increasing from a baseline mean of 34.6 (SD = 8.8) at T_0_ to 46.4 (SD = 4.3) at T_1_ (*p* < 0.001). The scores remained elevated at the one-month follow-up (T_2_) with a mean of 44.7 (SD = 4.9), significantly higher than the baseline (*p* < 0.001). However, the T_2_ scores were slightly but significantly lower than the immediate post-training scores at T_1_ (*p* = 0.018). In addition, self-efficacy scores increased substantially, from a mean of 26.2 (SD = 0.60) at T_1_ to 28.0 (SD = 0.62) at the one-month follow-up (T_2_) (*p* = 0.009) (see Figure 1 and Figure 2).

Figure 3 presents a scatterplot showing the relationship between change in GSES scores from T_1_ to T_2_ and the change in ECCC scores from T_0_ to T_1_. The Pearson correlation coefficient was −0.08 (*p* = 0.471), indicating no significant correlation between the two variables. The widely dispersed data points suggest considerable variability in how changes in self-efficacy correspond to changes in emergency care competency.

## 4. Discussion

This study employed a single-group repeated measures design to investigate the impact of experiential learning on nurse practitioners’ and nurse practitioner trainees’ performance in a three-minute emergency care simulation and their self-efficacy levels. Overall, the results indicate that experiential learning significantly improved both emergency care competency and self-efficacy. The participants’ mean ECCC scores showed a marked increase from the baseline (T_0_) to the immediate post-training assessment (T_1_) (*p* < 0.001) and remained elevated at the one-month follow-up (T_2_) (*p* < 0.001), though slightly lower than T_1_ (*p* = 0.018). This suggests that the skills acquired training were largely retained after one month, indicating the potential durability of the intervention.

Our findings are consistent with those of previous studies. For example, a quasi-experimental study of 86 nursing students found that a web-based experiential learning program significantly improved evidence-based practice knowledge, skills, and confidence in clinical questioning compared to traditional learning [26]. Similarly, experiential learning through simulations has been shown to enhance clinical judgment and performance among 144 first clinical semester, prelicensure, baccalaureate nursing students [27].

In addition to improved emergency care competency, our study showed a significant increase in self-efficacy, with GSES scores rising from T_1_ to T_2_ (*p* = 0.009). This finding aligns with prior research showing that simulation-based and experiential learning exercises can effectively enhance self-efficacy in emergency care settings. For example, a study of emergency nurses found that computer-based simulations significantly improved general self-efficacy and management skills [28]. Similarly, interprofessional simulation-based team training significantly improved confidence, self-efficacy in interprofessional communication, and emergency medicine skills among newly graduated doctors, nurses, auxiliary nurses, and students [29]. Moreover, an educational workshop was found to significantly improve nurses’ self-efficacy. The improvement was sustained after a 3-month follow-up period [30]. Collectively, these findings, along with our results, show the importance of self-efficacy in enhancing nurse practitioners’ confidence and readiness in emergency situations.

However, despite the significant improvements in both ECCC and GSES scores, our study found no significant correlation between the two (r = −0.08; *p* = 0.471). This contrasts with some studies, such as a cross-sectional study in Saudi Arabia that reported a positive correlation between nurses’ self-efficacy and their knowledge of cardiopulmonary resuscitation and defibrillation (r = 0.207; *p* < 0.001) [31]. Nevertheless, it should be noted that the cross-sectional study design precluded the establishment of the direction of the causal relationship.

While self-efficacy is often regarded as a key factor in improving clinical competencies, evidence suggests that this relationship is not always straightforward. For instance, some studies have demonstrated increases in self-efficacy without corresponding gains in clinical competency. A study on nursing students found that those who received self-efficacy prebriefing had significantly higher self-efficacy and clinical competency compared to a control group; however, no significant correlation between self-efficacy and clinical competency was observed (*p* = 0.207) [32]. Similarly, a systematic review reported that virtual reality simulations improved self-efficacy but did not result in significant improvements in nursing competency [33]. These findings, alongside our study results, indicate that while self-efficacy is an important and desirable outcome of training, it may not always directly translate into enhanced clinical performance. This suggests the need for training programs to address both self-efficacy and measurable clinical skills to ensure comprehensive professional development.

Several factors may influence both competency and self-efficacy, including experience, workplace environment, education level, and personal traits such as emotional intelligence and critical thinking [34,35]. Given the multifaceted nature of these relationships, future research should explore additional factors that contribute to both self-efficacy and clinical competence, as well as the potential mediating variables that may affect this complex interaction.

Several limitations of this study should be noted. First, the absence of a control group limits our ability to attribute the observed improvements solely to the intervention. Second, the study was conducted at a single regional hospital in southern Taiwan, which may restrict the generalizability of the findings to other nurse practitioner populations or settings. Third, the reliance on self-reported measures, rather than objective assessments, may not fully capture actual changes in performance or decision-making in real-world scenarios. Fourth, the ECCC is an unvalidated internal tool specifically developed for this study. Although the ECCC was designed to reflect key competencies needed in our acute care setting and was reviewed by emergency care experts for content relevance, it has not undergone formal psychometric validation. This limits the generalizability of the findings and suggests that further research should include validation of the ECCC to enhance its reliability and applicability in other settings. Lastly, this study did not examine long-term retention of emergency care skills or self-efficacy beyond the one-month follow-up.

Despite these limitations, this study provides valuable preliminary evidence on the potential of experiential learning to improve nurse practitioners’ emergency care competency and self-efficacy. Future research should address these limitations by including control groups and examining long-term outcomes to further validate the efficacy of experiential learning in preparing nurse practitioners for critical care situations.

## 5. Conclusions

The results of this study showed that experiential learning-based emergency care training significantly improved participants’ performance on a three-minute emergency simulation. Self-efficacy also improved from baseline to immediate post-training and remained elevated at the one-month follow-up. Throughout the training, participants strengthened their practical skills and confidence through simulation experiences, self-reflection, and instructor feedback. These findings support that experiential learning can enhance nurse practitioners’ responsiveness and confidence in emergency situations. Future research should explore the application of experiential learning in diverse clinical settings to further enhance both emergency care competencies and self-efficacy among nurse practitioners, ultimately contributing to improved healthcare outcomes.

## Figures and Tables

**Figure 1 healthcare-12-02333-f001:**
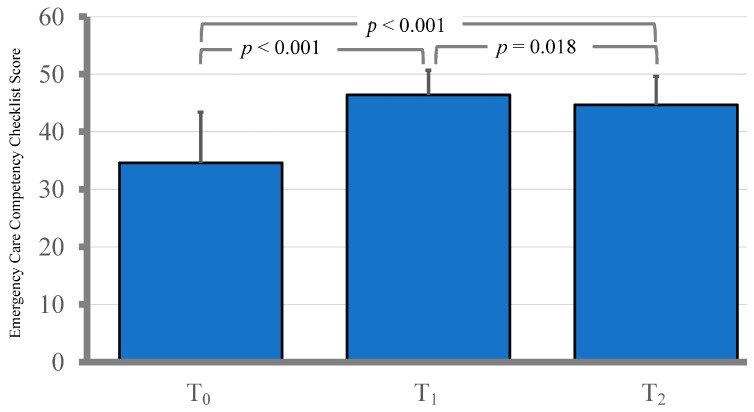
Emergency competency of participants measured by the Emergency Care Competency Checklist (ECCC) at T_0_, T_1_, and T_2_. *p*-values shown were adjusted using Bonferroni correction.

**Figure 2 healthcare-12-02333-f002:**
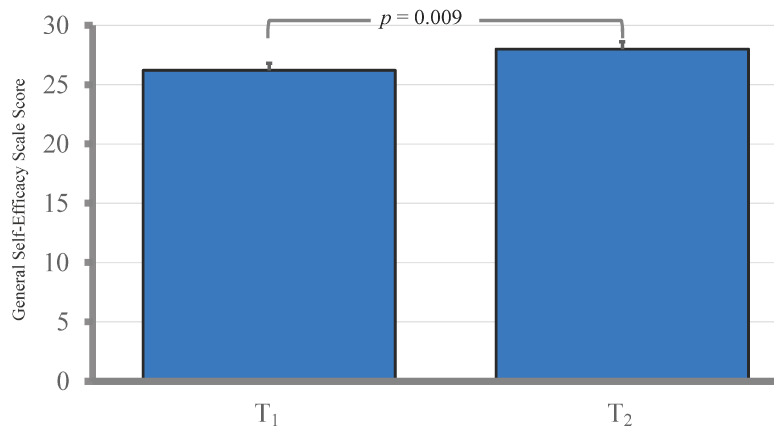
General Self-Efficacy Scale (GSES) score of participants at T_1_ and T_2_.

**Figure 3 healthcare-12-02333-f003:**
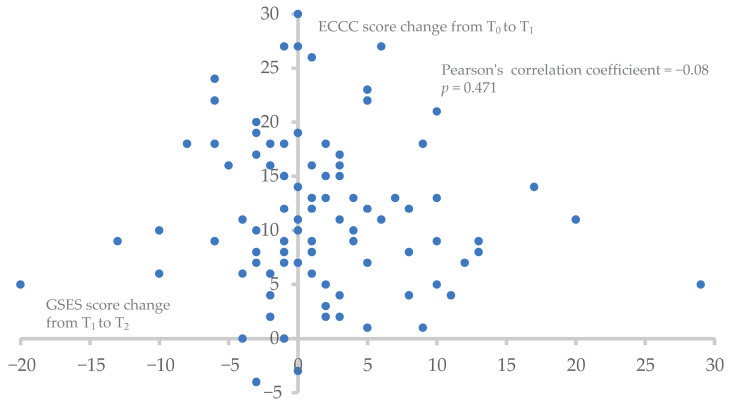
A scatterplot of the change in General Self-Efficacy Scale (GSES) scores from T_1_ to T_2_ versus the change in Emergency Care Competency Checklist (ECCC) scores from T_0_ to T_1_.

**Table 1 healthcare-12-02333-t001:** Basic characteristics of the study participants (n = 95).

Variable	Mean (SD) or n (%)
Age	42.1 (6.7)
Sex	
Male	4 (4.2)
Female	91 (95.8)
Work experience, years	19.1 (6.4)
Job title	
Nurse practitioner	89 (93.7)
Nurse practitioner trainee	6 (6.3)
Work unit	
Internal medicine	31 (32.6)
Surgery	30 (31.6)
Critical care	17 (17.9)
Other	17 (17.9)
Educational level	
Bachelor’s degree	68 (71.6)
Graduate degree	27 (28.4)
Professional level	
N (nurse practitioner trainee)	6 (6.3)
NP I (novice)	6 (6.3)
NP II (advanced beginner)	63 (66.3)
NP III (competent)	18 (18.9)
NP IV (proficient)	1 (1.1)
NP V (expert)	1 (1.1)
Holds emergency-care-training-related certificate	
Yes	90 (94.7)
No	5 (5.3)
Holds Advanced Cardiac Life Support (ACLS) certificate	
Yes	89 (93.7)
No	6 (6.3)
Holds Advanced Neonatal Life Support (ANLS) certificate	
Yes	3 (3.2)
No	92 (96.8)
Holds Advanced Pediatric Life Support (APLS) certificate	
Yes	1 (1.1)
No	94 (98.9)
Holds Neonatal Resuscitation Program (NRP) certificate	
Yes	9 (9.5)
No	86 (90.5)
Holds Emergency Trauma Technician Certificate (ETTC) certificate	
Yes	3 (3.2)
No	92 (96.8)

SD: standard deviation.

## Data Availability

The data presented in this study are available on request from the corresponding author.

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
