# Peer review of "Enhancing Nurse Practitioners’ Emergency Care Competency and Self-Efficacy Through Experiential Learning: A Single-Group Repeated Measures Studyâ€"

_healthcare, 2024, doi:10.3390/healthcare12232333_

Round 1
Reviewer 1 Report
Comments and Suggestions for Authors
Summary
This manuscript details a cohort of nurse practitioners and nurse practitioner students that received training to improve emergency response. Efficacy of the intervention was evaluated using standardized tools at three time points, pre-intervention, immediately post-intervention, and at one month post-intervention. The data provided demonstrates improved emergency performance capability and improved self-perception of efficacy in the immediate post-intervention time which was somewhat retained at one month post-intervention.
Abstract
The abstract describes the importance, method, outcomes, and potential application for further research clearly. However, I believe it would benefit from cutting the non-significant findings, which can still be presented but are detractors in this section of the manuscript.
Additionally:
22-23: please clarify what “91 being female” is referring to – is this the number of participants?
25: the presentation of the two p values would be stronger if it was contained in a separate sentence from the mean scores for the three time periods.
Introduction
Provides some background on nurse practitioner role in emergency situations, but primarily focuses on learning through experiential learning theory and increasing confidence to promote prompt application of knowledge. However, this section should also include demonstration of test validation and explanation of the choice to internally create the "Emergency Care Competency Checklist" (ECCC) and use the "General Self-Efficacy Scale" (GSES).
Additionally:
40: would benefit from examples of critical situations lead by nurse practitioners as frontline responders
44-45: refers to “these frameworks” but there is no framework introduced, only skills, confidence, and concepts. This leads to a lack of clarity.
63-72: data would be strengthened if study type was included to demonstrate veracity of outcomes
74-78: these statements require citations
81: in what settings are nurse practitioners preceding emergency response teams in patient care? This is not clear in any section of the paper and needs to be explained, especially as it the level of care expected from nurse practitioners differs between countries and even facilities.
Materials and Methods
Methods explained but difficult to follow. Test information on validity should be present in background section. Though, since the ECCC was created for this project the steps to create it could stay in this section.
Additionally:
98-114: this area needs to be reworked to provide a smooth timeline with clear progression and processes. Currently it is choppy and difficult to follow with doubling back on the timeline.
127-129: this approval should be at the top of the Methods section
132-133: why was the ECCC developed by physicians without input from nurse practitioners to validate scope of practice?
141-146: demonstrates that the "Emergency Care Competency Checklist" (ECCC) is a unvalidated internal test. This should be clarified from the beginning along with the rational for creating and using an unvalidated test in this setting.
Results
Subject characteristics and results of ECCC and GSES sores are reported and demonstrated on simplistic graphs that succeed in showing relationships and significance.
Additionally:
174 & 178: raw number and percentage should be presented
177: what is NPII level? Levels present but not explained in Table 1 (they are not universal and thus require additional information if you plan to present).
It is unclear why the majority of the nurse practitioners hold ACLS certificates and are only using an AED in the project. This level of performance is far below ACLS standard for a provider.
Discussion
While this section does discuss the results presented, it also diverges to background information supporting the concept of the intervention multiple times. While it is appropriate to discuss the findings of this study as similar findings to others, multiple paragraphs provide too much detail of other studies rather than demonstrating adherence to expected outcomes.
Additionally:
211: though there was an increase in competency and efficacy, they were not correlative in their outcomes. This should be specified.
216: the Park et al. 2020 study does not support the findings of this project, it demonstrates a similar outcome.
222-227: this statement was not proven by the project and should be cited or removed.
228: repeated drilling was not evaluated in this project
233-246: this is background information that supports your intervention trial – the paragraph needs to be reworked to be in the discussion section.
256-266: this is not analysis it is background
Conclusion
Solid summation of the study and results. Feels odd that the creation of the ECCC outcome measurement tool is not addressed in the discussion or the conclusion. Did the study require the creation of a tool? Was it a limitation to have a physician created and unvalidated tool?
Additionally:
This section should include that further research is needed regarding longer longitudinal retention, as this was presented as a limitation several times in the discussion.
Comments on the Quality of English Language
Please review for cohesiveness and flow.
Author Response
Reviewer 1, Comment #1:
Summary
This manuscript details a cohort of nurse practitioners and nurse practitioner students that received training to improve emergency response. Efficacy of the intervention was evaluated using standardized tools at three time points, pre-intervention, immediately post-intervention, and at one month post-intervention. The data provided demonstrates improved emergency performance capability and improved self-perception of efficacy in the immediate post-intervention time which was somewhat retained at one month post-intervention.
Response to Reviewer 1, Comment #1:
We appreciate Reviewer 1 for highlighting the key aspects of our study and providing a thorough review of our manuscript.
---------------------------------------------------------------------------------------
Reviewer 1, Comment #2:
Abstract
The abstract describes the importance, method, outcomes, and potential application for further research clearly. However, I believe it would benefit from cutting the non-significant findings, which can still be presented but are detractors in this section of the manuscript.
Response to Reviewer 1, Comment #2:
We appreciate the reviewer for the valuable feedback on the abstract. In response, we have revised the abstract to remove details of the non-significant findings, specifically the sentence: “However, no significant correlation was found between changes in ECCC and GSES scores (r = −0.08, p = 0.471).”
---------------------------------------------------------------------------------------
Reviewer 1, Comment #3:
22-23: please clarify what “91 being female” is referring to – is this the number of participants?
Response to Reviewer 1, Comment #3:
We thank reviewer 1‘s comment regarding clarification of the demographic data. In response, we have revised the sentence in the abstract to more clearly indicate that “91” refers to the number of participants who were female. The revised sentence now reads: “The mean age of participants was 42.1 years (SD = 6.7), with 91 out of 95 participants (95.8%) being female.” (line 22-23)
---------------------------------------------------------------------------------------
Reviewer 1, Comment #4:
25: the presentation of the two p values would be stronger if it was contained in a separate sentence from the mean scores for the three time periods.
Response to Reviewer 1, Comment #4:
In response, we have revised the original sentence into three separate sentences to improve clarity, as follows: “ECCC scores increased significantly from a baseline mean of 34.6 (standard deviation [SD] = 8.8 at T0 to 46.4 (SD = 4.3) at T1 (p < 0.001). Scores remained elevated at T2, with a mean of 44.7 (SD = 4.9), which was significantly higher than T0 (p < 0.001). However, scores at T2 were slightly lower than at T1 (p = 0.018).” (line 23-26)
---------------------------------------------------------------------------------------
Reviewer 1, Comment #5:
Introduction
Provides some background on nurse practitioner role in emergency situations, but primarily focuses on learning through experiential learning theory and increasing confidence to promote prompt application of knowledge. However, this section should also include demonstration of test validation and explanation of the choice to internally create the "Emergency Care Competency Checklist" (ECCC) and use the "General Self-Efficacy Scale" (GSES).
Response to Reviewer 1, Comment #5:
We appreciate the reviewer’s valuable comment. In response, we have added a detailed explanation of the rationale behind the development of the ECCC in Section 2.2 (Measurement Scales) rather than in the Introduction, as we believe this placement improves the flow of the manuscript. Given that the primary aim of the study is not scale development, including this information in the Methods section ensures a more focused Introduction. Specifically, the ECCC was designed through a series of collaborative meetings with attending physicians and emergency department attending physicians at the study hospital. The checklist’s content and structure were guided by a quality management analysis of emergency response incidents within our hospital, as well as the 2020 American Heart Association Guidelines Update for Cardiopulmonary Resuscitation and Emergency Cardiovascular Care. It encompasses essential emergency care competencies alongside common clinical errors observed in our setting, aiming to reflects both critical competencies and the practical challenges of emergency care.
In response to the comment regarding the rationale for selecting the General Self-Efficacy Scale (GSES), we have provided additional details in the manuscript (lines 169 to 174) to clarify our choice of the GSES. Specifically, we chose the GSES due to its strong psychometric properties, extensive cross-cultural validation, and suitability for assessing general self-efficacy in diverse contexts. The GSES has demonstrated reliability and criterion-related validity across a variety of populations, including Chinese samples, which aligns well with the goals and target population of our study.
---------------------------------------------------------------------------------------
Reviewer 1, Comment #6:
40: would benefit from examples of critical situations lead by nurse practitioners as frontline responders
Response to Reviewer 1, Comment #6:
We have added the following information according to the reviewer’s suggestion: "In Taiwan, regional and local hospitals often experience a shortage of resident physicians, who are typically concentrated in medical centers. As a result, nurse practitioners frequently serve as first-line responders in inpatient wards during night shifts or weekends, performing initial assessments when immediate medical support is limited. In these settings, when a patient’s condition deteriorates or vital signs become unstable, the primary nursing staff first contacts a nurse practitioner, who conducts an initial assessment and stabilizes the patient before determining whether additional physician support is needed. If cardiac or respiratory arrest occurs, a hospital-wide code call will be initiate to activate the cardiopulmonary resuscitation team, ensuring that the full emergency response team arrives promptly. In addition, nurse practitioners can provide adjunctive care under a physician's guidance, including ordering and interpreting diagnostic tests and initiating basic treatment protocols. This scope of practice enables nurse practitioners to effectively manage critical situations as primary responders and to coordinate with the clinical team, especially in urgent situations when physicians are not immediately available." (line 39-53)
---------------------------------------------------------------------------------------
Reviewer 1, Comment #7:
44-45: refers to “these frameworks” but there is no framework introduced, only skills, confidence, and concepts. This leads to a lack of clarity.
Response to Reviewer 1, Comment #7:
We thank the reviewer for highlighting this point of clarification. In response, we have revised the sentence as follows: “This intersection of skill and confidence aligns with two fundamental concepts in medical education—experiential learning and self-efficacy [3]—providing a foundation for understanding and enhancing the performance of nurse practitioners in acute care settings.” (line 61-64)
---------------------------------------------------------------------------------------
Reviewer 1, Comment #8:
63-72: data would be strengthened if study type was included to demonstrate veracity of outcomes
Response to Reviewer 1, Comment #8:
We have added the study type of each study, as follows: “Experiential learning has been shown to significantly enhance self-efficacy. In one study using a pre-test/post-test comparison group design, Doctor of Physical Therapy students who participated in pediatric experiential learning demonstrated improvements in both clinical reasoning and self-efficacy [9]. Similarly, a quasi-experimental study with a pre- and post-test design involving 249 nursing students and 50 medical students found that experiential learning in a mental health module improved their empathy and confidence in managing dangerous, aggressive, and violent patients [10]. Another quasi-experimental study, involving 285 nursing students who participated in simulated practice programs, showed significantly improvements in self-confidence for emergency intervention, which translated into enhanced knowledge and practical skills [11]. A single-group pre- and post-comparison study of pediatric emergency physicians also reported greater confidence and perceived ability to manage patients in a disaster situation following experiential learning through simulation-based work-shops [12].” (line 81-93)
---------------------------------------------------------------------------------------
Reviewer 1, Comment #9:
74-78: these statements require citations
Response to Reviewer 1, Comment #9:
We appreciate the reviewer’s feedback. The paragraph on lines 71–83 was intended to provide a summary of the studies cited above rather than introduce new information. In response, we have removed this paragraph and replaced it with a concise summary statement, which reads: "Collectively, these studies support the effectiveness of experiential learning in enhancing self-efficacy and confidence across diverse clinical settings, supporting its value as a powerful educational approach for improving healthcare professionals' readiness to manage complex and dynamic healthcare environments." (line 93-97)
---------------------------------------------------------------------------------------
Reviewer 1, Comment #10:
81: in what settings are nurse practitioners preceding emergency response teams in patient care? This is not clear in any section of the paper and needs to be explained, especially as it the level of care expected from nurse practitioners differs between countries and even facilities.
Response to Reviewer 1, Comment #10:
We have added the following information in response to the reviewer’s suggestion: "In Taiwan, regional and local hospitals often experience a shortage of resident physicians, who are typically concentrated in medical centers. As a result, nurse practitioners frequently serve as first-line responders in inpatient wards during night shifts or weekends, performing initial assessments when immediate medical support is limited. In these settings, when a patient’s condition deteriorates or vital signs become unstable, the primary nursing staff first contacts a nurse practitioner, who conducts an initial assessment and stabilizes the patient before determining whether additional physician support is needed. If cardiac or respiratory arrest occurs, a hospital-wide code call will be initiate to activate the cardiopulmonary resuscitation team, ensuring that the full emergency response team arrives promptly. In addition, nurse practitioners can provide adjunctive care under a physician's guidance, including ordering and interpreting diagnostic tests and initiating basic treatment protocols. This scope of practice enables nurse practitioners to effectively manage critical situations as primary responders and to coordinate with the clinical team, especially in urgent situations when physicians are not immediately available." (line 39-53)
---------------------------------------------------------------------------------------
Reviewer 1, Comment #11:
Materials and Methods
Methods explained but difficult to follow. Test information on validity should be present in background section. Though, since the ECCC was created for this project the steps to create it could stay in this section.
Response to Reviewer 1, Comment #11:
We have divided the original Section 2.2 into five distinct sections to improve clarity and organization, as follows:
2.2. Testing Sessions
2.3. Baseline and Post-Intervention Procedures
2.4. Assessment Tools and Scenarios
2.5. Examiner and Standardized Nurse Training
2.6. Measurement Scales
We hope this revision enhances the readability of the Methods section.
---------------------------------------------------------------------------------------
Reviewer 1, Comment #12:
98-114: this area needs to be reworked to provide a smooth timeline with clear progression and processes. Currently it is choppy and difficult to follow with doubling back on the timeline.
Response to Reviewer 1, Comment #12:
We have divided the original Section 2.2 into five distinct sections, as follows:
2.2. Testing Sessions
2.3. Baseline and Post-Intervention Procedures
2.4. Assessment Tools and Scenarios
2.5. Examiner and Standardized Nurse Training
2.6. Measurement Scales
We hope this revision enhances the readability of the Methods section.
---------------------------------------------------------------------------------------
Reviewer 1, Comment #13:
127-129: this approval should be at the top of the Methods section
Response to Reviewer 1, Comment #13:
We have moved the approval statement to the top of the Methods section.
---------------------------------------------------------------------------------------
Reviewer 1, Comment #14:
132-133: why was the ECCC developed by physicians without input from nurse practitioners to validate scope of practice?
Response to Reviewer 1, Comment #14:
We appreciate the reviewer for raising this point. The Emergency Care Competency Checklist (ECCC) was designed by teaching attending physicians with specific expertise in emergency care, ensuring that the checklist met the high standards required for accurate assessment in acute scenarios. The ECCC was modeled after Objective Structured Clinical Examination (OSCE)-style assessment tools and tailored to reflect essential tasks identified in our improvement initiatives within the hospital.
---------------------------------------------------------------------------------------
Reviewer 1, Comment #15:
141-146: demonstrates that the "Emergency Care Competency Checklist" (ECCC) is a unvalidated internal test. This should be clarified from the beginning along with the rational for creating and using an unvalidated test in this setting.
Response to Reviewer 1, Comment #15:
We acknowledge that the ECCC is an internally developed tool, designed specifically for this study to meet the unique needs of our institutional setting. Unlike traditional standardized questionnaires, the ECCC was created as an Objective Structured Clinical Examination (OSCE)-style assessment checklist, tailored to address specific competencies and areas for improvement identified within our hospital.
Although the ECCC has not undergone formal psychometric validation, it has been reviewed by a panel of emergency care experts to ensure content validity, aligning with the clinical and training requirements of our setting. We have now clarified in the discussion section with the following limitation: “Fourth, the ECCC is an unvalidated internal tool specifically developed for this study. Although the ECCC was designed to reflect key competencies needed in our acute care setting and was reviewed by emergency care experts for content relevance, it has not undergone formal psychometric validation. This limits the generalizability of the findings and suggests that further research should include validation of the ECCC to enhance its reliability and applicability in other settings.” (line 308-313)
---------------------------------------------------------------------------------------
Reviewer 1, Comment #16:
Results
Subject characteristics and results of ECCC and GSES sores are reported and demonstrated on simplistic graphs that succeed in showing relationships and significance.
Response to Reviewer 1, Comment #16:
We appreciate the reviewer’s positive feedback on the presentation of subject characteristics and the results of the ECCC and GSES scores.
---------------------------------------------------------------------------------------
Reviewer 1, Comment #17:
174 & 178: raw number and percentage should be presented
Response to Reviewer 1, Comment #17:
We have rewritten the paragraph and added raw number to the percentage as suggested.
---------------------------------------------------------------------------------------
Reviewer 1, Comment #18:
177: what is NPII level? Levels present but not explained in Table 1 (they are not universal and thus require additional information if you plan to present).
Response to Reviewer 1, Comment #18:
We have added the following description in the Results section: “In terms of skill level, 63 participants (66.3%) were at the NPII level. In Taiwan, nurse practitioners advance through a clinical ladder and certification system that supports career progression based on increasing clinical competence and responsibility. Starting at the trainee level, N represents nurse practitioner trainees who are preparing for clinical roles. From there, the levels range from NPI to NPV as follows: (1) NPI - Novice: Becoming familiar with the nurse practitioner scope of practice; (2) NPII - Advanced Beginner: Capable of handling basic nurse practitioner duties and providing individualized patient care; (3) NPIII - Competent: Able to develop evidence-based care plans and contribute opinions within the medical team; (4) NPIV - Proficient: Manages overall patient conditions, applies research to care, and coordinates with the medical team; (5) NPV - Expert: Makes rapid, accurate clinical judgments, integrates research, and influences the medical team and institution. (see Table 1).” (line 209-220)
---------------------------------------------------------------------------------------
Reviewer 1, Comment #19:
It is unclear why the majority of the nurse practitioners hold ACLS certificates and are only using an AED in the project. This level of performance is far below ACLS standard for a provider.
Response to Reviewer 1, Comment #19:
We thank the reviewer for this observation. The focus on AED use in our project was based on the latest evidence indicating that AEDs are the most effective initial intervention for cardiac arrest in the first few minutes. Given that nurse practitioners are often the first responders on scene within this critical window, we prioritized training on accurate and prompt AED use to optimize patient outcomes until the rest of the medical team arrives. This approach aligns with evidence-based practices for early intervention in emergency situations.
---------------------------------------------------------------------------------------
Reviewer 1, Comment #20:
Discussion
While this section does discuss the results presented, it also diverges to background information supporting the concept of the intervention multiple times. While it is appropriate to discuss the findings of this study as similar findings to others, multiple paragraphs provide too much detail of other studies rather than demonstrating adherence to expected outcomes.
Response to Reviewer 1, Comment #20:
In response to the reviewer’s response, we have revised and shorten the section to focus more directly on interpreting our study’s findings in relation to similar studies, rather than providing detailed descriptions of other research.
---------------------------------------------------------------------------------------
Reviewer 1, Comment #21:
211: though there was an increase in competency and efficacy, they were not correlative in their outcomes. This should be specified.
Response to Reviewer 1, Comment #21:
We thank the reviewer for the comment. We have already addressed the lack of correlation between competency and self-efficacy outcomes in both the Results and Discussion sections. In the Results, Figure 3 illustrates this relationship, with a Pearson correlation coefficient of −0.08 (p = 0.471), indicating no significant correlation. Moreover, in the Discussion, we note that while self-efficacy is often thought to improve clinical competencies, other studies have also shown this relationship is not straightforward. Our findings align with previous research, suggesting that increases in self-efficacy do not necessarily translate into improved clinical performance.
---------------------------------------------------------------------------------------
Reviewer 1, Comment #22:
216: the Park et al. 2020 study does not support the findings of this project, it demonstrates a similar outcome.
Response to Reviewer 1, Comment #22:
In response to the reviewer's comment, we have clarified that the studies demonstrate similar outcomes rather than directly supporting our study's findings by revising the first sentence of the paragraph to: “Our findings are consistent with those of previous studies.” (line 257)
---------------------------------------------------------------------------------------
Reviewer 1, Comment #23:
222-227: this statement was not proven by the project and should be cited or removed.
Response to Reviewer 1, Comment #23:
We have removed the paragraph as suggested by the reviewer.
---------------------------------------------------------------------------------------
Reviewer 1, Comment #24:
228: repeated drilling was not evaluated in this project
Response to Reviewer 1, Comment #24:
We have removed the sentence.
---------------------------------------------------------------------------------------
Reviewer 1, Comment #25:
233-246: this is background information that supports your intervention trial – the paragraph needs to be reworked to be in the discussion section.
Response to Reviewer 1, Comment #25:
We have reworked the paragraph to emphasize how our findings are consistent with and contribute to the broader literature on self-efficacy and experiential learning, as follows: “In addition to improved emergency care competency, our study showed a significant increase in self-efficacy, with GSES scores rising from T1 to T2 (p = 0.009). This finding aligns with prior research showing that simulation-based and experiential learning exercises can effectively enhance self-efficacy in emergency care settings. For example, a study of emergency nurses found that computer-based simulations significantly improved general self-efficacy and management skills [27]. Similarly, interprofessional simulation-based team training significantly improved confidence, self-efficacy in interprofessional communication, and emergency medicine skills among newly graduated doctors, nurses, auxiliary nurses, and students [28]. Moreover, educational workshop was found to significantly improve nurses' self-efficacy. The improvement was sustained after a 3-month follow-up period [29]. Collectively, these findings, along with our results, show the importance of self-efficacy in enhancing nurse practitioners' confidence and readiness in emergency situations.” (line 263-274)
---------------------------------------------------------------------------------------
Reviewer 1, Comment #26:
256-266: this is not analysis it is background
Response to Reviewer 1, Comment #26:
We have reworded the paragraph as follows: “While self-efficacy is often regarded as a key factor in improving clinical competencies, evidence suggests that this relationship is not always straightforward. For in-stance, some studies have demonstrated increases in self-efficacy without corresponding gains in clinical competency. A study on nursing students found that those who received self-efficacy prebriefing had significantly higher self-efficacy and clinical competency compared to a control group; however, no significant correlation between self-efficacy and clinical competency was observed (p = 0.207) [31]. Similarly, a systematic review reported that virtual reality simulations improved self-efficacy but did not result in significant improvements in nursing competency [32]. These findings, alongside our study results, indicate that while self-efficacy is an important and desirable outcome of training, it may not always directly translate into enhanced clinical performance. This suggests the need for training programs to address both self-efficacy and measurable clinical skills to ensure comprehensive professional development.” (line 283-295)
---------------------------------------------------------------------------------------
Reviewer 1, Comment #27:
Conclusion
Solid summation of the study and results. Feels odd that the creation of the ECCC outcome measurement tool is not addressed in the discussion or the conclusion. Did the study require the creation of a tool? Was it a limitation to have a physician created and unvalidated tool?
Response to Reviewer 1, Comment #27:
As mentioned in our response to Comment #15, the ECCC is an internally developed tool, designed specifically for this study to meet the unique needs of our institutional setting. Unlike traditional standardized questionnaires, the ECCC was created as an Objective Structured Clinical Examination (OSCE)-style assessment checklist, tailored to address specific competencies and areas for improvement identified within our hospital. We have mentioned in the limitation that the ECCC has not undergone formal psychometric validation, limiting the generalizability of the findings.
---------------------------------------------------------------------------------------
Reviewer 1, Comment #28:
This section should include that further research is needed regarding longer longitudinal retention, as this was presented as a limitation several times in the discussion.
Response to Reviewer 1, Comment #28:
We have indicated that in one of the study limitations is “the study did not examine long-term retention of emergency care skills or self-efficacy beyond the one-month follow-up.” (line 313-314) and we suggested that “Future research should address these limitations by including control groups and examining long-term outcomes to further validate the efficacy of experiential learning in preparing nurse practitioners for critical care situations.” (317-319)
---------------------------------------------------------------------------------------
Reviewer 1, Comment #29:
Please review for cohesiveness and flow.
Response to Reviewer 1, Comment #29:
We appreciate the review for the thorough review of our manuscript and for the valuable feedback. We have reviewed the manuscript to improve cohesiveness and flow throughout. We have restructured certain paragraphs and streamlined information to enhance readability.
Reviewer 2 Report
Comments and Suggestions for Authors
Congratulations on the work developed, we list some aspects that could be subject of some reflection on your part.
In relation to the introduction, the allusion to Patricia Banner's model from insider to expert might make sense.
Maybe you can analyzed the time of experience to understand if it had importance.
Author Response
Reviewer 2, Comment #1:
Congratulations on the work developed, we list some aspects that could be subject of some reflection on your part.
Response to Reviewer 2, Comment #1:
We thank reviewer 2 for the positive feedback and for taking the time to review our work.
---------------------------------------------------------------------------------------
Reviewer 2, Comment #2:
In relation to the introduction, the allusion to Patricia Banner's model from insider to expert might make sense.
Response to Reviewer 2, Comment #2:
We appreciate the reviewer’s insightful suggestion regarding the inclusion of Patricia Benner’s Novice to Expert model in the introduction. In response, we have added the following text to the introduction section: “This progression of clinical competence can be explained by Patricia Benner’s novice to expert model, which describes five levels of skill acquisition in nursing practice: novice, advanced beginner, competent, proficient, and expert [2]. Benner’s model provides a theoretical foundation for understanding the development of nurse practitioners as they gain experience, allowing them to handle complex scenarios with greater autonomy and confidence.” (line 56-61) We have also included a new reference #2 to support this addition. We believe this addition strengthens the theoretical context of our manuscript.
---------------------------------------------------------------------------------------
Reviewer 2, Comment #3:
Maybe you can analyzed the time of experience to understand if it had importance.
Response to Reviewer 2, Comment #3:
We thank the reviewer for this suggestion. In response, we conducted an additional analysis using "work experience" (in years) as a variable, categorizing participants into two groups based on the median value of 20 years. We performed a repeated measures ANOVA with the binary work experience variable as an independent variable, and the results indicated that work experience was not a significant factor (p = 0.178 for ECCC and p = 0.697 for GSES). The conclusions of our study remain unchanged from the original analysis. We appreciate your suggestion, which allowed us to further confirm the robustness of our findings.
Reviewer 3 Report
Comments and Suggestions for Authors
Introduction: The article makes a significant contribution to the literature on emergency training for nurses. By focusing on self-efficacy and practical skills, the study offers an important perspective for improving educational practices and can guide health training policies.
Methodology: The absence of a control group limits conclusions about the causality of the observed effects. The diversity of contexts would increase external validity. An appendix detailing the items of each scale used is suggested.
Results: Presented clearly, with tables and figures that make it easy to understand the changes in competence and self-efficacy scores over time. Statistical analysis is appropriate, using paired t-tests and Bonferroni corrections for multiple comparisons.
Discussion: Further analysis of the practical implications for nursing education and hospital training policies is suggested.
Conclusion: Adequately summarizes the main findings and reinforces the importance of experiential learning in developing nurses' competence and confidence.
Author Response
Reviewer 3, Comment #1:
Introduction: The article makes a significant contribution to the literature on emergency training for nurses. By focusing on self-efficacy and practical skills, the study offers an important perspective for improving educational practices and can guide health training policies.
Response to Reviewer 3, Comment #1:
We greatly appreciate for the reviewer’s positive feedback. The encouraging comments reinforce the value of our work, and we remain committed to furthering research in this area.
---------------------------------------------------------------------------------------
Reviewer 3, Comment #2:
Methodology: The absence of a control group limits conclusions about the causality of the observed effects. The diversity of contexts would increase external validity. An appendix detailing the items of each scale used is suggested.
Response to Reviewer 3, Comment #2:
We agree that the absence of a control group limits the ability to draw causal inferences from the observed effects. This limitation has been acknowledged in the discussion section of the manuscript, as follows: ”the absence of a control group limits our ability to attribute the observed improvements solely to the intervention.” (line 302-303)
In addition, our appendix to the manuscript contains the items of the ECCC.
---------------------------------------------------------------------------------------
Reviewer 3, Comment #3:
Results: Presented clearly, with tables and figures that make it easy to understand the changes in competence and self-efficacy scores over time. Statistical analysis is appropriate, using paired t-tests and Bonferroni corrections for multiple comparisons.
Response to Reviewer 3, Comment #3:
We thank the reviewer for the positive feedback on the clarity and presentation of the results.
---------------------------------------------------------------------------------------
Reviewer 3, Comment #4:
Discussion: Further analysis of the practical implications for nursing education and hospital training policies is suggested.
Response to Reviewer 3, Comment #4:
We thank the reviewer for the thoughtful suggestion. We acknowledge that we could have highlighted how experiential learning can be integrated into nursing curricula to enhance self-efficacy and clinical competence and discussed the potential for scaling such training approaches in hospital settings to improve emergency response outcomes. However, we believe that before conducting a long-term randomized controlled trial to further validate these findings, it may be premature to speculate extensively on hospital training policies. We appreciate the suggestion and recognize the importance of this analysis for future research directions.
---------------------------------------------------------------------------------------
Reviewer 3, Comment #5:
Conclusion: Adequately summarizes the main findings and reinforces the importance of experiential learning in developing nurses' competence and confidence.
Response to Reviewer 3, Comment #5:
We appreciate the review and positive feedback on the conclusion of our manuscript.